# Modulation of CD8^+^ T Cell Responses by Radiotherapy—Current Evidence and Rationale for Combination with Immune Checkpoint Inhibitors

**DOI:** 10.3390/ijms242316691

**Published:** 2023-11-24

**Authors:** Seung Hyuck Jeon, Changhoon Song, Keun-Yong Eom, In Ah Kim, Jae-Sung Kim

**Affiliations:** Department of Radiation Oncology, Seoul National University Bundang Hospital, Seoul National University College of Medicine, Seongnam 13620, Republic of Korea; hyck9004@naver.com (S.H.J.); hasadim2@naver.com (C.S.); 978sarang@snubh.org (K.-Y.E.); inah@snubh.org (I.A.K.)

**Keywords:** radiotherapy, CD8^+^ T cells, immune checkpoint inhibitors

## Abstract

Radiotherapy for cancer has been known to affect the responses of immune cells, especially those of CD8^+^ T cells that play a pivotal role in anti-tumor immunity. Clinical success of immune checkpoint inhibitors led to an increasing interest in the ability of radiation to modulate CD8^+^ T cell responses. Recent studies that carefully analyzed CD8^+^ T cell responses following radiotherapy suggest the beneficial roles of radiotherapy on anti-tumor immunity. In addition, numerous clinical trials to evaluate the efficacy of combining radiotherapy with immune checkpoint inhibitors are currently undergoing. In this review, we summarize the current status of knowledge regarding the changes in CD8^+^ T cells following radiotherapy from various preclinical and clinical studies. Furthermore, key biological mechanisms that underlie such modulation, including both direct and indirect effects, are described. Lastly, we discuss the current evidence and essential considerations for harnessing radiotherapy as a combination partner for immune checkpoint inhibitors.

## 1. Introduction

Radiotherapy (RT) is one of the major pillars of anti-cancer therapies and is used for the treatment of patients with various cancer types. Radiation brings about a cascade of molecular events within tumor cells, primarily by causing DNA double-strand breaks [1], that ultimately result in the mitotic death of the cells. Harnessing the mechanisms of direct tumor cell killing, approximately 60% of patients with cancer, including newly diagnosed and recurrent cancer, undergo RT with the aim of eradicating the tumor, preventing relapses, or relieving symptoms induced by tumors.

In addition to the ability of RT to achieve local control of the irradiated tumor, dozens of cases with response in unirradiated tumors, which is called an abscopal effect, have been reported to date [2]. First described in 1953 [3], the abscopal effect is now understood as an immune cell-mediated phenomenon; following the study that reported the difference in tumor susceptibility to RT between T-cell-competent and T-cell-depleted mice [4], a number of studies showed a direct linkage between the abscopal effect and mechanisms involving immune cells [5,6,7]. Although the abscopal effect after RT without any systemic treatment is rarely seen, the advent of immune checkpoint inhibitors (ICIs), which amplify anti-tumor immune responses by blocking negative immune regulators such as PD-1 and CTLA-4, has significantly heightened interest in the synergy between RT and ICIs, due to the capacity of RT to induce anti-tumor immune responses.

Among the myriad of immune cell populations that participate in anti-tumor or pro-tumor immune responses, CD8^+^ T cells are one of the most extensively investigated populations in the context of cancer-related immunity. Furthermore, CD8^+^ T cells play pivotal roles in driving the anti-tumor effects of ICIs, as evidenced by both preclinical and clinical studies [8,9]. As part of the adaptive immune system, CD8^+^ T cells specialize in recognizing specific peptide epitopes loaded on the major histocompatibility complex (MHC) class I molecules. CD8^+^ T cells are a heterogeneous population with a diverse repertoire, and recent analyses utilizing single-cell sequencing technologies have introduced deeper insights into the complexity of CD8^+^ T cells.

Given the importance of CD8^+^ T cells in anti-tumor immunity, a growing number of preclinical and clinical studies have explored the portraits of CD8^+^ T cell responses induced by RT. In this review, we first briefly address the key features and roles of CD8^+^ T cells in cancer. Then, we describe how CD8^+^ T cells respond to RT based on the evidence from animal studies and cancer patients. We also describe the mechanisms that underlie the responses of CD8^+^ T cells upon RT. Based on the evidence, we discuss how the alteration of CD8^+^ T cells induced by RT may be harnessed in the era of immunotherapy.

## 2. Features and Roles of CD8^+^ T Cells in Cancer

CD8^+^ T cells play crucial roles in anti-tumor immunity, owing to the cytoplasmic localization of a majority of neoantigens and tumor-associated antigens, collectively referred to as tumor antigens. Because of the universal expression pattern of MHC class I molecules, tumor-specific CD8^+^ T cells can recognize epitopes loaded on both tumor cells and antigen-presenting cells; meanwhile, some tumor cells lose MHC class I molecules on their surface, assisting tumor cells to evade elimination by CD8^+^ T cells [10]. Type 1 conventional dendritic cells (cDC1s) are responsible for priming tumor-specific CD8^+^ T cells at tumor-draining lymph nodes (TDLNs) [11]. The stimulated tumor-specific CD8^+^ T cells enter the circulation, infiltrate into the tumor microenvironment, and encounter tumor cells. Upon meeting tumor cells, CD8^+^ T cells secrete proinflammatory cytokines, including interferon-γ, and cytolytic enzymes, such as granzymes and perforin, which induce apoptosis of the target cells.

CD8^+^ T cells in cancer show distinct characteristics compared to those in acute infection. During the acute phase, naïve CD8^+^ T cells are differentiated into effector CD8+ T cells that can directly kill target cells. Following the clearance of antigens, the majority of effector T cells die, and some of them persist to become memory CD8^+^ T cells, consisting of effector memory T cells that migrate to peripheral tissues for immediate effector function and central memory T cells that home to secondary lymphoid organs for differentiation into effector cells upon antigenic challenge [12]. However, CD8^+^ T cells are in an environment where antigens are not cleared, deviating the differentiation of CD8^+^ T cells away from the classical pathway. The persistence of tumor cells leads to continuous T cell receptor (TCR) signaling within tumor-specific CD8^+^ T cells, ultimately rendering these cells dysfunctional via a process termed exhaustion. Exhausted CD8^+^ T cells are characterized by the expression of multiple inhibitory immune checkpoint receptors, such as PD-1, Tim-3, Lag-3, and TIGIT, and significant impairment in their effector functions [13]. The exhaustion process cannot be reversed due to their epigenetically fixed status [14]. Despite their compromised function, the exhausted CD8^+^ T cells contribute to the anti-tumor activity within the tumor [15]. Indeed, an increase in the exhausted CD8^+^ T cells following ICIs is associated with better clinical response [16].

Recent advances in single-cell profiling technology have allowed us to observe the detailed portraits of CD8^+^ tumor-infiltrating lymphocytes (TIL). For example, CD8^+^ TILs with interferon-stimulated gene signatures or stress response genes and those with NK-like features, including expression of killer cell immunoglobulin-like receptors, have been identified [17,18]. This subset of CD8^+^ T cells produces IL-17 that promotes the exhaustion of CD8^+^ TILs and subsequent tumor progression [19]. A subpopulation among CD8^+^ T cells with immunosuppressive roles has been reported in autoimmune and infectious diseases [20], but their existence and functions in the context of cancer have not been investigated.

## 3. Changes in CD8^+^ T Cells following RT

### 3.1. Evidence for CD8^+^ T Cell Responses via RT

CD8^+^ T cells are considered to be primarily responsible for the abscopal effect of RT. The activation of circulating CD8^+^ T cells and infiltration of CD8^+^ T cells in the tumor microenvironment of tumor-bearing mice were increased by local RT [21,22,23,24,25,26,27,28]. Importantly, CD8^+^ T cell population-specific tumor antigens were expanded by local RT in murine tumor models [26,29,30,31]. Moreover, local RT also increases the cytokine and cytolytic enzyme production of CD8^+^ TILs [24,30,31]. Accordingly, in vivo depletion of CD8^+^ T cells was found to hamper the anti-tumor effects of RT and preclude the occurrence of the abscopal effect [29,32,33,34].

Studies using clinical samples also support the activation of CD8^+^ T cells, possibly including tumor-specific cells, by RT. Chow et al. convincingly described the activation of anti-tumor CD8^+^ T cells by RT [35]. They harnessed blood and tissue samples of patients with renal cell carcinoma undergoing preoperative stereotactic body radiotherapy (SBRT) with 15 Gy in a single fraction. Using TCR sequencing of CD8^+^ T cells from the tumor and peripheral blood, the authors showed the increase in tumor-enriched clonotypes among circulating CD8^+^ T cell pool after irradiation, suggesting that systemic anti-tumor responses are induced by local RT in the immunologically hot tumor.

Responses of circulating CD8^+^ T cells following SBRT in early-stage non-small cell lung cancer (NSCLC) were evaluated in two studies. Zhang et al. analyzed six patients with stage I NSCLC undergoing SBRT with a daily dose of 6 Gy or 8 Gy and showed that the proportion of CD8^+^ T cells among immune cells and the production of proinflammatory cytokines by CD8^+^ T cells after in vitro stimulation was significantly increased by SBRT [36]. Similarly, a study by Gkika et al. that examined 50 early-stage NSCLC patients treated with an ablative dose of SBRT demonstrated that expression levels of Ki-67 and IFN-γ in CD8^+^ T cells were significantly increased following SBRT, although the absolute count of CD8^+^ T cells was diminished [37]. Notably, these changes were observed only in patients who were treated with SBRT with a daily dose of 10 Gy or lower. Kim et al. analyzed the circulating CD8^+^ T cells from patients with locally advanced NSCLC and found that the proliferation of circulating CD8^+^ T cells and relative frequency of CD39^+^ tumor-specific cells among circulating CD8^+^ T cells were increased by concurrent chemoradiotherapy (CCRT) [38]. Analyses on tumor samples of NSCLC also showed that CD8^+^ TILs were also increased following CCRT [39,40,41].

Activation of CD8^+^ T cells by RT was also observed in patients with localized or metastatic prostate cancer. In a study by Evans et al., SBRT to oligometastatic lesions of prostate cancer with various dose–fractionation schedules with a daily dose of 10 Gy or higher increased tumor-reactive CD8^+^ T cells, defined by the expression of CD11a in the peripheral blood [42]. Similarly, another study showed that circulating CD8^+^ T cells reactive to tumor-associated antigens were increased in two prostate cancer patients at 1 month after RT [43]. Palermo et al. compared the effects of varying RT schemes, including SBRT, hypofractionated RT, and conventionally fractionated RT, and revealed that the proportion of CD8^+^ T cells among circulating T cells was increased after RT regardless of the RT scheme [44]. Notably, effector memory CD8^+^ T cells and VISTA-expressing CD8^+^ T cells were increased only in patients undergoing SBRT. Interestingly, Hoffmann et al. demonstrated that the decrease in circulating CD45RA^+^CD28^−^ effector CD8^+^ T cells after conventionally fractionated RT was correlated with a higher dose to pelvic bone marrow or blood vessels [45], indicating that both daily dose and irradiated site may be associated with the responses of CD8^+^ T cells following RT. In contrast, the infiltration of CD8^+^ T cells in prostatic tumors was decreased 2 weeks after SBRT to the prostate in a study of six patients [46].

Explorations in cervical cancer also provide insights into how CD8^+^ T cells respond to RT. Studies have shown that CCRT induces increased clonality [47] and a proportion of HPV-specific clonotypes [48] among circulating T cells. A study by Li et al. showed that PD-1^+^ CD8^+^ T cells, which are known to be enriched for tumor-specific cells, are expanded in circulation following CCRT [49], while another study demonstrated a decrease in the PD-1^+^ CD8^+^ T cell population [47]. On the other hand, the infiltration of CD8^+^ T cells within cervical cancer was significantly decreased after CCRT [47,50,51]. Additionally, expressions of negative immune checkpoint receptors on CD8^+^ TILs were decreased [52], while those of activation markers were significantly increased following CCRT [50]. Chen et al. dissected the heterogeneity of CD8^+^ T cell infiltration by CCRT and showed that the CCRT-mediated increase in CD8^+^ TILs is associated with intratumoral induction of interferons [53]. Interestingly, naive CD8^+^ T cells in TDLNs were diminished by CCRT, and CCR5^+^ CXCR3^+^ CD8^+^ T cells in TDLNs were increased after a lower total dose (39.6 Gy) but not after a higher dose (50 Gy) [54].

In rectal cancer, where neoadjuvant CCRT is provided as a standard-of-care treatment, the density of CD8^+^ TILs was increased after CCRT [55,56,57,58,59,60,61,62,63,64]. Interestingly, the clonal composition of TILs became significantly diverse after CCRT [65]. Similarly, the infiltration of CD8^+^ T cells into tumors was increased following CCRT in esophageal cancer [66,67,68], and responses of circulating T cells to tumor-associated antigens were enhanced following CCRT to esophageal cancer [69]. Increased infiltration of CD8^+^ T cells following RT was also noted in patients with sarcoma [70,71]. Analysis of peripheral blood from metastatic breast cancer undergoing SBRT to bone metastasis showed that circulating PD-1^+^ CD8^+^ T cells were activated following SBRT [72]. In contrast, the amount of stromal CD8^+^ TILs was decreased following RT in squamous cell carcinoma of the oral cavity [73]. Moreover, neoadjuvant CCRT reduced the infiltration of CD8^+^ T cells in clinical T4 gastric cancer [74].

In summary, a majority of the reported data suggests that local RT induces both systemic activation and local expansion of anti-tumor CD8^+^ T cells. However, some reports examining tumor tissues imply the negative impact of RT on CD8^+^ T cell responses in the local tumor microenvironment. The studies on CD8^+^ T cell responses using peripheral blood samples and tumor tissues from cancer patients are listed in Table 1 and Table 2, respectively.

### 3.2. Factors That Influence RT-Induced CD8^+^ T Cell-Responses

Multiple tumor- or treatment-related factors are considered to be associated with the T-cell responses. First of all, total dose or fraction size may be a crucial factor, as shown by the study of the murine breast cancer model demonstrating decreased immunostimulatory effects in a dose above 12 Gy [75]. Consistent with this concept, a study on early-stage NSCLC patients demonstrated that the proliferation of circulating CD8^+^ T cells was increased by SBRT with a daily dose of 10 Gy or less but not with higher doses [37]. Nonetheless, an analysis of prostate cancer patients and metastatic breast cancer patients did not observe any difference among varying daily doses [44,72], suggesting that the relationship between CD8^+^ T cell responses and the daily dose should be investigated in detail.

Furthermore, various chemotherapeutic agents also have the capacity to modulate immune cells. The majority of aforementioned studies, especially those that quantified the density of CD8^+^ TILs, investigated the changes in CD8^+^ T cells by CCRT. Various cytotoxic chemotherapeutic drugs and targeted agents are capable of not only inducing anti-tumor CD8^+^ T cell responses but also depleting CD8^+^ T cells via their direct effects [76]. Since it is impossible to dissect the changes in CD8^+^ T cells into those resulting from RT and systemic therapies, the studies that analyzed the patients receiving CCRT should be interpreted with extreme caution.

The treatment field is also considered to have an impact on the T-cell responses. Anti-PD-1 blockade is known to provide limited efficacy for metastatic cancer to the liver [77,78] or bone [79] owing to the distinct microenvironment. Hence, the irradiated organ or site may also contribute to the T-cell responses following RT. McGee et al. reported that activated memory CD8^+^ T cells in peripheral blood were expanded by SBRT to parenchymal metastases but not to brain or bone metastases [80]. Moreover, multiple lines of preclinical evidence suggest that irradiating TDLN has detrimental effects in terms of systemic T cell responses and abscopal effects [30,31,81,82], possibly owing to the lymphodepleting effects of RT. Hence, intentional or unintentional irradiation of regional lymphatics in cancer patients may also affect the responses of CD8^+^ T cells.

The type of irradiation might also affect the responses of CD8^+^ T cells. While most of the studies used X-rays for irradiation, few studies have explored the effects of particle beams, such as protons and carbon ions, on the activity of CD8^+^ T cells. Proton therapy has been shown to elicit CD8^+^ T cell responses, and its effect was similar to X-ray irradiation [83,84]. Abscopal response in a sarcoma patient undergoing proton therapy has also been reported [85]. Interestingly, carbon ion irradiation has shown an enhanced ability to upregulate calreticulin and increase CD8^+^ TILs compared to X-ray [86,87]. Despite the lack of comprehensive analyses on the differential effects on CD8^+^ T cell responses according to the type of irradiation, several clinical trials involving particle beam therapy and ICIs are ongoing [88], and their results are strongly awaited.

Given the complex interaction between various components of the tumor microenvironment, responses of cells other than CD8^+^ T cells also determine the responses of CD8^+^ T cells upon RT. It has been shown that regulatory CD4^+^ T cells (T_REG_ cells) are activated and become more suppressive following RT, both in murine models and clinical samples [72,89]. Additionally, infiltration of tumor-associated macrophages (TAMs) is increased by RT by chemokines and other soluble proteins, including TGF-β and colony-stimulating factor 1 [90]. Polarization of TAMs into either anti-tumor or pro-tumor phenotype by RT seems to highly depend on the context, possibly including the RT dose [91]. RT is also capable of modulating myeloid-derived suppressor cells (MDSCs), although their changes are inconsistently reported across the studies. These immune cells (i.e., T_REG_ cells, TAMs, and MDSCs) are able to inhibit the anti-tumor functions of CD8^+^ T cells in the tumor microenvironment. Consistently, the depletion of those immunosuppressive cells improved the efficacy of local RT in multiple preclinical models [28,92]. Moreover, recent evidence suggests that cancer-associated fibroblasts are involved in immune responses in tumor microenvironments. Nicolas et al. reported that inflammatory fibroblasts induced by IL-1 inhibit the infiltration of CD8^+^ T cells, resulting in resistance to chemoradiotherapy in rectal cancer [93]. The responses of the diverse cell populations by RT are, in general, conflicting between the studies and should be carefully deconvoluted in future works.

Last but not least, tumor types and their characteristics are involved in the efficacy of RT to elicit anti-tumor CD8^+^ T cell responses. For instance, the amount of CD8^+^ TILs was decreased in cervical cancer but increased in rectal cancer following CCRT in most studies, as described above. Moreover, studies using clinical samples are mostly skewed to cancer types that include RT as a definitive or neoadjuvant setting, although some scrutinized the changes in T cells during palliative RT. Therefore, the results of individual studies should be interpreted cautiously and not be generalized into other disease or treatment settings.

## 4. Mechanisms of RT-Induced T-Cell Responses

### 4.1. Activation of Innate Immune Cells

Innate immune cells play crucial roles in orchestrating the anti-tumor functions of CD8^+^ T cells. In the tumor microenvironment, local RT triggers a cascade of responses to modulate innate immune cells. RT induces DNA-sensing pathways, such as the cGAS-STING pathway, in host or tumor cells to upregulate the production of type I interferon [75,94,95], which in turn activates tumor-infiltrating dendritic cells (DC) to augment their activity to cross-prime tumor-specific CD8^+^ T cells [95]. Gupta et al. observed that irradiation increases the expression of costimulatory molecules, including CD70 and CD86, on tumor-infiltrating DCs [29]. Consequently, infiltration of functional CD8^+^ T cells into the tumor microenvironment is increased after local RT [22]. These findings from animal models are also supported by the increase in plasma level of type I interferons in patients after a combination of local RT and ICIs [96]. Vanpouille-Box et al. showed that a single ablative dose (>12 Gy) induces *Trex1* that degrades cytosolic DNA molecules, preventing the activation of the cGAS-STING pathway, the release of type I interferons, and even abscopal effects [75]. Supporting these findings, the addition of STING agonist to local RT improves tumor control, which is accompanied by activation of CD8^+^ T cells [94,97]. These results indicate that the activation of DCs plays an important role in promoting CD8^+^ T cell responses after RT.

The release of soluble factors that stimulate innate immune cells also contributes to the responses of CD8^+^ T cells. Danger-associated molecular patterns, such as high-mobility group protein B1 (HMGB1), calreticulin, and ATP, are released following a process called immunogenic cell death [98]. These molecules induce the maturation of DCs, leading to their recruitment and activation. For instance, HMGB1 interacts with Toll-like receptor 4 (TLR4) expressed on the surface of DCs, which subsequently activates NF-κB signaling pathway to promote the activation of DCs [99]. TLR4-deficient mice exhibited a remarkable decrease in the infiltration of tumor-specific CD8^+^ T cells [100]. Additionally, calreticulin exposed to the cell surface promotes phagocytosis of the dying cells [101], rendering innate immune cells to activate CD8^+^ T cells more proficiently. Indeed, there are multiple translational studies that combined local RT and TLR agonists to enhance the activation of DCs [102,103]. Moreover, several clinical trials incorporating TLR agonists and RT are ongoing [104].

RT also generates reactive oxygen species and reactive nitrogen species that are capable of recruiting neutrophils and NK cells [105], which interact with anti-tumor CD8^+^ T cells [106]. Moreover, RT-induced pyroptosis improves the antigen-presenting function of DCs and infiltration of CD8^+^ T cells, although the exact mechanisms remain to be clarified [107]. Further studies are needed to gain further insights into various mechanisms underpinning responses of anti-tumor CD8^+^ T cells mediated by innate immune cells.

### 4.2. Generation and Release of Tumor Antigens

Tumor antigens include neoantigens that are generated by somatic nonsynonymous mutation and tumor-associated antigens that are overexpressed in tumor cells compared to normal cells. Expression and immunogenicity of tumor antigens are essential for responses of CD8^+^ T cells, as evidenced by the increased response rate to ICIs in tumors with high mutational burden [108]. Changes in the TCR repertoire of tumor-infiltrating T cells following local RT indicate that the antigenicity of tumor cells can be modulated by RT [109,110,111].

Irradiation elevated the intracellular protein levels and, subsequently, the activity of the transporter associated with antigen processing, which is involved in the cross-presentation for CD8^+^ T cells [112]. Of note, irradiation generated a novel set of MHC class I-binding peptides. Upregulation of immunoproteasome subunits, which are involved in the production of peptides with higher binding capacity to MHC class I molecules than standard proteasome, may partly explain the increased neoepitopes following RT [113]. Lhuillier et al. reported direct evidence that the expression of immunogenic neoantigens on murine breast cancer cells is increased by RT [114]. Interestingly, Lussier et al. showed that irradiation induces novel mutations that generate immunogenic epitopes for CD8^+^ T cells in the murine sarcoma model [115]. Upregulation of tumor antigens following irradiation in vitro or in vivo has also been reported in other preclinical studies [83,96]. Importantly, the expression of tumor-associated antigens, including NY-ESO-1 and CT7, was increased after local RT in patients with sarcoma [70]. Although multiple layers of evidence suggest that tumor antigens are upregulated by RT, the underlying mechanisms remain to be revealed.

### 4.3. Increased Susceptibility of Tumor Cells to CD8^+^ T Cell-Mediated Death

The expression of MHC class I molecules on tumor cells is required for CD8^+^ T cells to kill the tumor cells. Although MHC class I molecules are expressed on all normal nucleated cells, some cancer cells do not express MHC class I molecules to evade anti-tumor immune reaction [10]. Notably, studies using animal tumor models demonstrated that RT upregulates MHC class I molecules on tumor cells [21,113,116] as a result of increased degradation of cytosolic proteins [112]. Similarly, RT with or without concurrent chemotherapy to cancer patients significantly increased the expression of MHC class I molecules in cancer patients [70]. The mTOR-signaling pathway and NF-κB signaling pathway are responsible for the RT-induced upregulation of MHC class I molecules [113].

Irradiation also enhances the expression of Fas (CD95) on tumor cells [105,114]. When Fas binds to the FasL, which is typically expressed on CD8^+^ T cells, it activates a cascade of reactions involving caspase that ultimately leads to apoptosis. The cytolytic activity of tumor-specific CD8^+^ T cells was decreased when Fas:FasL interaction was blocked [117].

### 4.4. Recruitment and Retention of CD8^+^ T Cells

Tumor cells and various tumor-infiltrating immune cells secrete a series of chemokines to recruit immune cells that shape the tumor microenvironment. CD8^+^ T cells also harness various chemokine receptors expressed on their cell surface to navigate to and remain at the tumor site [118]. Several studies proposed that local RT upregulates the production of chemokines to increase the infiltration of CD8^+^ T cells. Matsumura et al. showed that irradiation upregulates the expression of CXCL16 that recruits CXCR6^+^ CD8^+^ T cells in a murine breast cancer model [119]. Importantly, infiltration of CD8^+^ T cells was decreased in CXCR6-deficient mice with impaired efficacy of local RT. Another study showed that RT-induced production of CXCL10 by lung cancer cells activated CXCR3^+^ CD8^+^ T cells [120]. Additionally, the expression of CCL22 in nasopharyngeal cancer xenograft models was upregulated by RT, which was associated with the increased infiltration of CD8^+^ T cells [121]. Enhanced production of other chemokines, including CCL2 [122,123,124], CCL5 [33], CXCL1 [122], and CXCL9 [125], by tumor or normal cells has been reported, yet their roles in the regulation of CD8^+^ T cell trafficking are unknown.

### 4.5. Direct Induction of Apoptosis

Hematopoietic cells, including CD8^+^ T cells, are susceptible to radiation-induced cell death. Indeed, the number of CD8^+^ TILs shortly after RT is reduced, probably via direct effects of radiation, and then restored afterward [126]. Interestingly, the sensitivity of CD8^+^ T cells to radiation differs according to their differentiation status. Tabi et al. showed that naïve and early memory T cells are more sensitive to irradiation-induced apoptosis compared to CD45RA^–^ T cells [43]. Moreover, CD8^+^ TILs are more radioresistant than CD8^+^ T cells at the circulation or lymphoid tissues and survive following RT to exert anti-tumor activity in a murine tumor model [127]. Since the discrimination between pre-existing and newly infiltrated CD8^+^ T cells is difficult in human samples, whether tumor-specific CD8^+^ TILs resist RT and maintain their anti-tumor functions remains unanswered.

## 5. Combination of RT and ICIs

ICIs exert anti-tumor effects mainly by reinvigorating the functions of exhausted CD8^+^ T cells and recruiting novel tumor-specific CD8^+^ T cells from the periphery [128,129]. Based on the aforementioned evidence demonstrating the ability of local RT to provoke responses of CD8^+^ T cells, local RT has been regarded as a reasonable partner for combination with ICIs [130]. The co-administration of local RT and ICIs is considered tolerable, as reported by numerous clinical works [131,132,133]. The genetic deletion of PD-1 in mice significantly improved the systemic anti-tumor efficacy of RT [134]. Accordingly, multiple preclinical studies using mouse models revealed that RT synergistically augments the efficacy of anti-PD-1/PD-L1 blockade [21,24,135,136,137,138] or anti-CTLA-4 blockade [34,109,139]. Anti-tumor effects of the combination strategies were abolished after in vivo depletion of CD8^+^ [135,136,140], supporting the CD8^+^ T cell-mediated mechanisms underpinning the synergism.

Unsurprisingly, CD8^+^ T cells play a crucial role in the combination strategies of RT and ICIs in cancer patients. The activation of circulating CD8^+^ T cells was observed in patients with metastatic melanoma, showing abscopal response via the combination therapy [141,142]. In a study dissecting the clonal composition of CD8^+^ T cells in a metastatic melanoma patient undergoing pembrolizumab and brain RT, a proportion of tumor-enriched clonotypes were expanded in peripheral blood after RT [143]. In the clinical trial of RT and anti-CTLA-4 blockade in NSCLC patients, the clonal expansion of circulating T cells was more prominent in responders compared to non-responders [96]. Additionally, the expansion of neoantigen-specific CD8^+^ T cells among circulating CD8^+^ T cells was observed. Of note, pembrolizumab plus SBRT increased the infiltration of CD8^+^ T cells, as well as those expressing CD103, more prominently than pembrolizumab alone in patients with metastatic NSCLC [144], suggesting that effects of RT on tumor-specific CD8^+^ T cells may enhance the anti-tumor activity of ICIs.

Recently, results of the clinical trials assessing the efficacy of adding RT to ICIs have been reported; however, the role of local RT is conflicting across the studies. A pooled analysis of two randomized trials showed that a combination of RT and pembrolizumab showed significantly improved survival outcomes compared to pembrolizumab alone in metastatic NSCLC patients [145]. In contrast, the addition of SBRT to anti-PD-1 blockade did not provide any benefit for metastatic head and neck squamous cell carcinoma [146] and adenoid cystic carcinoma [147]. Additionally, local RT did not improve the treatment outcomes when added to the co-administration of anti-PD-1/PD-L1 and anti-CTLA-4 antibodies in patients with NSCLC [148] and Merkel cell carcinoma [149]. These findings suggest that the incorporation of local RT alongside ICIs does not consistently provide beneficial outcomes.

The impact of local RT on the anti-tumor effects of ICIs may vary across cancer types, regimen of ICIs, and timing and dose–fractionation scheme of RT, which were mostly discussed in the previous section (see Section 3.2). Indeed, the responses of tumor-specific CD8^+^ T cells on ICIs are different according to the features of the tumor microenvironment, especially the exhaustion status of tumor-specific CD8^+^ T cells [16,150]. Moreover, anti-PD-1/PD-L1 and anti-CTLA-4 blocking antibodies elicit distinct cellular responses, not just in CD8^+^ T cells but also in CD4^+^ T cells, particularly T_REG_ cells [8], which directly regulate the functions of CD8^+^ TILs. Additionally, Wei et al. recently showed that administration of the first dose of anti-PD-1 antibodies prior to irradiation hampered the abscopal effects of local RT using a mouse tumor model [140]. This phenomenon was owing to the sensitization of CD8^+^ TILs to RT-induced DNA damage after anti-PD-1 blockade. Hence, optimizing the treatment setting for the meaningful benefit of local RT is necessary.

Modulation of other immune cells is also an important factor that determines the combined efficacy of ICIs and RT. A series of reports revealed that blocking the PD-1:PD-L1 pathway in T_REG_ cells enhances their suppressive capacity [151,152,153]. In addition, RT has been shown to increase the infiltration of T_REG_ cells into the tumor microenvironment and enhance their suppressive functions [89,154,155]. Thus, the anti-tumor activity of the combination strategy also depends on how T_REG_ cells are modulated by the therapies, although evidence regarding the association between treatment- or tumor-related factors and responses of T_REG_ cells is scarce. Moreover, phenotypes and functions of various immune cells, including TAMs, NK cells, and DCs, are altered by local RT as well as ICIs [156]. However, the impacts of the combination therapy on these immune cells have not been studied in depth. Therefore, further studies as to how the therapies reshape the tumor microenvironment and systemic immunity would provide profound insight regarding the optimal strategy to combine the local RT and ICIs.

## 6. Conclusions and Future Directions

Current evidence from preclinical and clinical research supports the modulation of local and systemic CD8^+^ T cell responses by local RT. The changes in anti-tumor CD8^+^ T cells and their mechanisms are summarized in Figure 1. How clinically relevant factors, such as dose–fractionation schedule, irradiation site, and characteristics of the irradiated tumor, impact the CD8^+^ T cell responses is largely unknown and remains to be elucidated. Moreover, the mechanisms behind the RT-induced CD8^+^ T cell responses are not fully understood. Recent techniques, including single-cell profiling, may facilitate the detailed characterization of CD8^+^ T cell responses and their relationship with key features of tumor or RT. Future works aiming to study the unraveled issues in this field will suggest the optimal RT strategies for evoking anti-tumor CD8^+^ T cell responses in the era of immunotherapy.

## Figures and Tables

**Figure 1 ijms-24-16691-f001:**
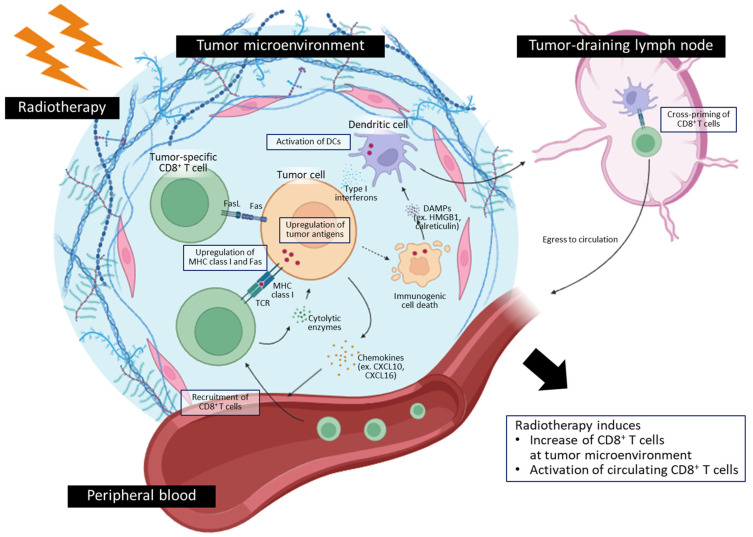
Summary for key features of radiotherapy-induced changes in anti-tumor CD8^+^ T cells and their underlying mechanisms.

**Table 1 ijms-24-16691-t001:** Studies regarding responses of circulating CD8^+^ T cells following radiotherapy in cancer patients.

Study	Cancer	RT	Systemic Therapy	Patients	Findings	Refs.
Chow et al. (2020)	RCC	15 Gy/1 fx	No	11	- Increase in tumor-enriched clonotypes at 2 weeks after RT	[35]
Zhang et al. (2017)	NSCLC (early stage)	48 Gy/6 fx, 48 Gy/8 fx	No	6	- Increase in inflammatory cytokine production potency at 3 weeks after RT	[36]
Gkika et al. (2023)	NSCLC (early stage)	45 Gy/3 fx, 50 Gy/5 fx, 60 Gy/8 fx, 66 Gy/12 fx	No	50	- Increase in Ki-67 expression after RT (≤10 Gy/fx) - Increase in IFN-γ production potency during and after RT	[37]
Kim et al. (2022)	NSCLC (locally advanced)	66 Gy/30 fx	Paclitaxel/cisplatin or Paclitaxel/carboplatin	24	- Increase in Ki-67 expression at the end of CCRT - Increase in PD-1^+^CD39^+^ tumor-specific CD8^+^ T cells at the end of CCRT	[38]
Evans et al. (2019)	Prostate ca (oligometastatic)	16–24 Gy/1 fx, 30 Gy/3 fx	Androgen deprivation therapy	37	- Decrease in CCR7^-^CD45RA^+^ effector CD8^+^ T cells after RT	[42]
Tabi et al. (2010)	Prostate ca (locally advanced)	55 Gy/20 fx	Androgen deprivation therapy	12	- Increase in TAA-specific CD8^+^ T cell responses after RT	[43]
Palermo et al. (2023)	Prostate ca	40 Gy/3 fx, 62 Gy/20 fx, 66–69 Gy/30 fx	Androgen deprivation therapy	18	- Increase in CCR7^+^CD45RA^−^ central memory CD8^+^ T cells during RT - Increase in CCR7^-^CD45RA^−^ effector memory CD8^+^ T cells after RT (40 Gy/3 fx)	[44]
Hoffmann et al. (2022)	Prostate ca (locally advanced)	70–78 Gy/ 35–39 fx	Androgen deprivation therapy	18	- Increase in Ki-67 expression during and after RT	[45]
Li et al. (2021)	Cervical ca (locally advanced)	50 Gy/25 fx + 25–36 Gy/5–6 fx	Cisplatin	55	- Decrease in PD-1^+^ CD8^+^ T cells during and after CCRT - Increase in clonality of T cells	[47]
Colbert et al. (2022)	HPV-positive ca (mostly cervical ca)	≥45 Gy/25 fx	Cisplatin	86	- Increase in HPV-specific T cells during CCRT	[48]
Li et al. (2021)	Cervical ca or Esophageal ca	50.4 Gy/28 fx + 30 Gy/5 fx (cervical ca) 60 Gy/30 fx (esophageal ca)	Cisplatin (cervical ca) Cisplatin/Docetaxel (esophageal ca)	57	- Increase in PD-1^hi^ CD8^+^ T cell during CCRT	[49]
Suzuki et al. (2012)	Esophageal ca	60–66 Gy/ 30–33 fx	Cisplatin-based	16	- Increase in TAA-specific CD8^+^ T cell responses	[69]
Jeon et al. (2023)	Breast ca (metastatic)	12–20 Gy/1 fx or 24–27 Gy/3 fx	Various (90%)	30	- Increase in Ki-67 expression on PD-1^+^ CD8^+^ T cells	[72]

**Table 2 ijms-24-16691-t002:** Studies regarding responses of tumor-infiltrating CD8^+^ T cells following radiotherapy in cancer patients.

Study	Cancer	RT	Systemic Therapy	Patients	Findings	Refs.
Yoneda et al. (2019)	NSCLC (locally advanced)	Median 60 Gy	Platinum doublet	23	- Increase in stromal CD8^+^ TIL density after CCRT	[39]
Choe et al. (2019)	NSCLC (locally advanced)	40–50 Gy	Docetaxel/carboplatin	43	- Increase in CD8^+^ TIL density after CCRT	[40]
Shirasawa et al. (2020)	NSCLC (locally advanced)	Mostly ≤66 Gy	Platinum doublet, Carboplatin	14	- Increase in tumoral CD8^+^ TIL density after CCRT	[41]
Kane et al. (2023)	Prostate cancer (high-risk)	24 Gy/3 fx	No	6	- Decrease in CD8^+^ TIL density after RT	[46]
Li et al. (2021)	Cervical ca (locally advanced)	50 Gy/25 fx + 25–36 Gy/5–6 fx	Cisplatin	55	- Decrease in CD8^+^ TIL density during and after CCRT	[47]
Dorta-Estremera et al. (2018)	Cervical ca (locally advanced)	40–45 Gy + brachytherapy	Cisplatin	20	- Decrease in CD8^+^ TIL proportion during CCRT - Increase in Ki-67 expression on CD8^+^ TIL during CCRT	[50]
Mori et al. (2021)	Cervical ca (locally advanced)	50 Gy/25 fx + 24 Gy/4 fx (m/c)	Cisplatin (64%)	75	- Decrease in stromal CD8^+^ TIL density during CCRT (no change in RT alone)	[51]
Herter et al. (2023)	Cervical ca (locally advanced)	Not documented	Cisplatin, Carboplatin	22	- Decrease in expressions of PD-1 and TIGIT on CD8^+^ TIL after CCRT	[52]
Chen et al. (2021)	Cervical ca (locally advanced)	46 Gy/23 fx + 24–32 Gy/4–5 fx	Cisplatin, Cisplatin/5-FU	30	- No change in CD8^+^ TIL density during CCRT - Increase in CD8^+^ TIL density correlated with nuclear IRF1 in tumor cells	[53]
Baeten et al. (2006)	Rectal ca (locally advanced)	50.4 Gy/28 fx, 25 Gy/5 fx	Capecitabine, 5-FU (for 50.4 Gy/28 fx)	32	- Increase in CD8^+^ TIL density after CCRT (25 Gy/5 fx)	[55]
Teng et al. (2015)	Rectal ca (locally advanced)	40–45 Gy/25–28 fx + 5.4 Gy boost	Fluoropyrimidine-based	103	- Increase in CD8^+^ TIL density after CCRT and RT alone	[56]
Teng et al. (2015)	Rectal ca (locally advanced)	40–45 Gy/25–28 fx + 5.4 Gy boost	Fluoropyrimidine-based	62	- Increase in CD8^+^ TIL density after CCRT	[57]
Lim et al. (2014)	Rectal ca (locally advanced)	45–50.4 Gy/25–28 fx, 25 Gy/5 fx	5-FU (long-course)	52	- Increase in CD8^+^ TIL density after CCRT (long-course)	[58]
Shinto et al. (2014)	Rectal ca (locally advanced)	20 Gy/5 fx	UFT	93	- Increase in stromal CD8^+^ TIL density after CCRT	[59]
Lim et al. (2017)	Rectal ca (locally advanced)	Median 50.4 Gy/28 fx	Capecitabine, 5-FU	123	- Increase in CD8^+^ TIL density after CCRT	[60]
Cho et al. (2022)	Rectal ca (locally advanced)	Median 50 Gy/25 fx	Capecitabine, 5-FU	101	- Increase in CD8^+^ TIL density after CCRT	[61]
Mirjolet et al. (2018)	Rectal ca (locally advanced)	N/D	Fluoropyrimidine-based	132	- Increase in CD8^+^ TIL/Treg ratio after CCRT	[62]
Ogura et al. (2018)	Rectal ca (locally advanced)	45 Gy/25 fx, 50.4 Gy/28 fx	Capecitabine	281	- Increase in stromal CD8^+^ TIL density after CCRT	[63]
Matsutani et al. (2018)	Rectal ca (locally advanced)	50.4 Gy/28 fx	Fluoropyrimidine-based	31	- Increase in CD8^+^ TIL density after CCRT	[64]
Akiyoshi et al. (2021)	Rectal ca (locally advanced)	45 Gy/25 fx, 50.4 Gy/28 fx	Capecitabine	65	- Increase in clonal diversity of TIL after CCRT (poor responders)	[65]
Kelly et al. (2018)	Esophageal ca (locally advanced)	41.4–54 Gy	N/D	31	- Increase in CD8^+^ TIL density after CCRT	[66]
Zhou et al. (2020)	Esophageal ca (locally advanced)	Median 40 Gy (36–46 Gy)	Cisplatin-based	82	- Increase in CD8^+^ TIL density after CCRT	[67]
Chen et al. (2022)	Esophageal ca (locally advanced)	N/D	N/D	23	- Increase in expression of CD8 in tumor after CCRT	[68]
Sharma et al. (2013)	Sarcoma	50 Gy (m/c)	No	38	- Increase in CD8^+^ TIL density after RT	[70]
Keung et al. (2018)	Undifferentiated pleomorphic sarcoma	50 Gy/25 fx	Yes (59%; m/c = adrimycin/ifosfamide)	17	- Increase in CD8^+^ TIL density after (CC)RT	[71]
Tabachnyk et al. (2012)	Oral cavity SqCC (locally advanced)	50.4 Gy/28 fx	Cisplatin/5-FU	22	- Decrease in stromal CD8^+^ TIL density after CCRT	[73]
Huang et al. (2018)	Gastric ca (locally advanced)	N/D	Paclitaxel/carboplatin	68	- Increase in CD8^+^ TIL density after CCRT (T2) - Decrease in CD8^+^ TIL density after CCRT (T4)	[74]

Abbreviation: TIL, tumor-infiltrating lymphocyte; N/D, not documented; m/c, most common.

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
