# Peer review of "Modulation of CD8+ T Cell Responses by Radiotherapy—Current Evidence and Rationale for Combination with Immune Checkpoint Inhibitors"

_ijms, 2023, doi:10.3390/ijms242316691_

Round 1

Reviewer 1 Report

Comments and Suggestions for Authors

This review provides a detailed summary of current combinations of radiotherapy (RT) and immune check-point inhibitors (ICIs). Overall, this paper is well organized and includes all the useful information. For better improvement, I pose two points for the consideration.

1. AS the authors said that “CD8+ T cells are one of the most extensively investigated population in the context of cancer-related immunity”, “Given the importance of CD8+ T cells in anti-tumor immunity, growing number of preclinical and clinical studies are exploring the portraits of CD8+ T cell responses induced by RT”, “phenotypes and functions of various immune cells, including tumor-associated macrophages, NK cells, and dendritic cells, are altered by local RT as well as ICIs”, I still want to know why other kinds of immune cells like CD4+ or dendritic cells gained the less attention. If there are some reports about other immune cells affected by RT, it is still to provide a table to list all of them.

2. Anti-PD-1/PD-L1 and anti-CTLA-4 blocking antibodies have been widely used in combination with RT. Is there reports about the combinations of RT and other small molecule activatiors for innate immunity like STING agonists? These activatiors also present the high potency in reactivating CD8+ T cells. If not, this aspect is necessary to be added in the future directions.

Author Response

This review provides a detailed summary of current combinations of radiotherapy (RT) and immune check-point inhibitors (ICIs). Overall, this paper is well organized and includes all the useful information. For better improvement, I pose two points for the consideration.

  1. AS the authors said that “CD8+ T cells are one of the most extensively investigated population in the context of cancer-related immunity”, “Given the importance of CD8+ T cells in anti-tumor immunity, growing number of preclinical and clinical studies are exploring the portraits of CD8+ T cell responses induced by RT”, “phenotypes and functions of various immune cells, including tumor-associated macrophages, NK cells, and dendritic cells, are altered by local RT as well as ICIs”, I still want to know why other kinds of immune cells like CD4+ or dendritic cells gained the less attention. If there are some reports about other immune cells affected by RT, it is still to provide a table to list all of them.

We agree that describing how radiotherapy alters various immune cells is crucial in that there is a complex crosstalk among immune cells, including CD8+ T cells, in the tumor microenvironment. Indeed, there are numerous published studies regarding the responses of various immune cells other than CD8+ T cells following RT. Moreover, the reported results are somewhat inconsistent among the studies. Therefore, we incorporated a brief description regarding the changes of other immune cells into the revised manuscript.

  1. Anti-PD-1/PD-L1 and anti-CTLA-4 blocking antibodies have been widely used in combination with RT. Is there reports about the combinations of RT and other small molecule activatiors for innate immunity like STING agonists? These activatiors also present the high potency in reactivating CD8+ T cells. If not, this aspect is necessary to be added in the future directions.

As the reviewer suggested, there are reports about combination of RT and small molecule that activates innate immune cells, including STING agonists. Activation of innate immunity by RT is substantially relevant to the responses of CD8+ T cells. In the revised manuscript, we introduced representative studies of STING agonists and TLR agonists in combination with RT.

Reviewer 2 Report

Comments and Suggestions for Authors

The major purpose of the manuscript presented by Jeon et al. was to summarize the current knowledge on the modulation of CD8+ T cell response by radiotherapy from various preclinical and clinical investigations. In more detail, the authors focus on mechanisms of a direct or indirect modulation of CD8-T cell activity by ionizing radiation and discuss evidence for CD8+ T cell modulation for harnessing radiotherapy as an additive in immune checkpoint inhibitor therapy.

In summary, there are no major issues regarding this manuscript, as the issue addressed is interesting and of clinical relevance, and the authors put high efforts into drafting the text. Overall, the review is well-designed and presented, supported by a clearly arranged figure and two tables. There are, however, some minor issues, as mentioned successive, aiming to further improve the significance of the content.

Minor points of criticism:

1.     Authors should  describe CD8+ cell subsets in more detail, as not all readers will be familiar with effector and effector memory  etc. subpopulations.

2.     In addition, other stromal TMA components, like fibroblasts, should be briefly included in the review in a separate paragraph. For instance, radiation-induced senescence of interleukin 1 (IL-1)-induced inflammatory fibroblasts is reported to hamper CD8+ TIL infiltration in rectal cancer by an excessive production of extracellular matrix, while antagonization of IL1 increases CD8-infiltration (Nicolas et al., Cancer Cell 2022;40:168).

3.     Authors introduced most acronyms in the text but should carefully read the manuscript, as some abbreviations e.g. SBRT, CCRT, TDLNs were not introduced at their first appearance.

Comments on the Quality of English Language

In summary, there are no major issues regarding this manuscript, as the issue addressed is interesting and of clinical relevance, and the authors put high efforts into drafting the text. Overall, the review is well-designed and presented, supported by a clearly arranged figure and two tables. There are, however, some minor issues, as mentioned successive, aiming to further improve the significance of the content. After performing these improvements, I do recommend the manuscript for publication in the International Journal of Molecular Sciences.

Author Response

The major purpose of the manuscript presented by Jeon et al. was to summarize the current knowledge on the modulation of CD8+ T cell response by radiotherapy from various preclinical and clinical investigations. In more detail, the authors focus on mechanisms of a direct or indirect modulation of CD8-T cell activity by ionizing radiation and discuss evidence for CD8+ T cell modulation for harnessing radiotherapy as an additive in immune checkpoint inhibitor therapy.

In summary, there are no major issues regarding this manuscript, as the issue addressed is interesting and of clinical relevance, and the authors put high efforts into drafting the text. Overall, the review is well-designed and presented, supported by a clearly arranged figure and two tables. There are, however, some minor issues, as mentioned successive, aiming to further improve the significance of the content.

Minor points of criticism:

  1. Authors should describe CD8+ cell subsets in more detail, as not all readers will be familiar with effector and effector memory  etc. subpopulations.

As the reviewer suggested, we agree that an introduction to the subsets of CD8+ T cells is needed to enhance readability for the readers. We added a brief overview regarding the differentiation and role of effector and memory (both effector memory and central memory) CD8+ T cells in the revised manuscript.

  1. In addition, other stromal TMA components, like fibroblasts, should be briefly included in the review in a separate paragraph. For instance, radiation-induced senescence of interleukin 1 (IL-1)-induced inflammatory fibroblasts is reported to hamper CD8+ TIL infiltration in rectal cancer by an excessive production of extracellular matrix, while antagonization of IL1 increases CD8-infiltration (Nicolas et al., Cancer Cell 2022;40:168).

We appreciate the insightful comment and fully agree that description for the role of cancer-associated fibroblasts on modulating CD8+ T cell responses is needed for our review. Accordingly, we included the recent paper suggested by the reviewer in the revised manuscript.

  1. Authors introduced most acronyms in the text but should carefully read the manuscript, as some abbreviations e.g. SBRT, CCRT, TDLNs were not introduced at their first appearance.

We apologize for the omission of descriptions for certain acronyms. As the reviewer pointed out, we carefully reviewed the revised manuscript and introduced all abbreviations at their first appearance.